# Homocysteine Attack on Vascular Endothelium—Old and New Features

**DOI:** 10.3390/ijms26136298

**Published:** 2025-06-30

**Authors:** Loredana Liliana Hurjui, Cristina Claudia Tarniceriu, Dragomir Nicolae Serban, Ludmila Lozneanu, Gabriela Bordeianu, Alin Horatiu Nedelcu, Alin Constantin Panzariu, Raluca Jipu, Ruxandra Maria Hurjui, Daniela Maria Tanase, Ionela Lacramioara Serban

**Affiliations:** 1Department of Morpho-Functional Sciences II, Physiology Discipline, “Grigore T. Popa” University of Medicine and Pharmacy, 700115 Iasi, Romania; loredana.hurjui@umfiasi.ro (L.L.H.); dragomir.serban@umfiasi.ro (D.N.S.); alin.pinzariu@umfiasi.ro (A.C.P.); raluca.jipu@umfiasi.ro (R.J.); ionela.serban@umfiasi.ro (I.L.S.); 2Hematology Laboratory, “Sf. Spiridon” County Clinical Emergency Hospital, 700111 Iasi, Romania; 3Department of Morpho-Functional Sciences I, Discipline of Anatomy, “Grigore T. Popa” University of Medicine and Pharmacy, 700115 Iasi, Romania; alin.nedelcu@umfiasi.ro; 4Hematology Clinic, “Sf. Spiridon” County Clinical Emergency Hospital, 700111 Iasi, Romania; 5Department of Morpho-Functional Sciences I, Discipline of Histology, “Grigore T. Popa” University of Medicine and Pharmacy, 700115 Iasi, Romania; ludmila.lozneanu@umfiasi.ro (L.L.); hurjui_ruxandra-maria@d.umfiasi.ro (R.M.H.); 6Department of Morpho-Functional Sciences II, Biochemistry Discipline, “Grigore T. Popa” University of Medicine and Pharmacy, 700115 Iasi, Romania; gabriela.bordeianu@umfiasi.ro; 7Department of Internal Medicine, “Grigore T. Popa” University of Medicine and Pharmacy, 700111 Iasi, Romania; daniela.tanase@umfiasi.ro; 8Clinic of Internal Medicine, “St. Spiridon” Emergency Hospital Iasi, 700111 Iasi, Romania

**Keywords:** homocysteine, endothelial dysfunction, hyperhomocysteinemia

## Abstract

Homocysteine (Hcy) is becoming a well-established risk factor for cardiovascular disease (CVD), mainly involving endothelial dysfunction and atherogenesis. Endothelial dysfunction is reflected primarily in the complex regulation of the main physiological and pathophysiological processes. There is increasing evidence regarding abnormally high concentrations of plasma total homocysteine, or plasma hyperhomocysteinemia, contributing to endothelial dysfunction, inflammation, and CVD. This clinical and experimental study examined the connection between Hcy and cardiovascular disease risk. Homocysteine is a marker of total vascular damage that must be monitored and controlled as early as possible. Dietary and lifestyle changes are recommended for most patients with hyperhomocysteinemia (Hhcy). The purpose of this paper is to review the data from the specialized literature that demonstrate that there is a direct link between endothelial injury and increased homocysteine levels, identifying existing evidence, describing new mechanisms, and exploring potential new therapeutic options. These aspects continue to be debated, and additional efforts are required to refine therapeutic strategies and to investigate the potential implications of Hcy in health and disease.

## 1. Introduction

Homocysteine (Hcy) is a methionine-derived, non-proteinogenic, sulfur-containing amino acid recognized for its role in various physiological and pathological processes. Plasma Hcy levels can be modulated by several factors, including genetic mutations, nutritional deficiencies, high methionine intake, underlying medical conditions (such as chronic renal failure, hypothyroidism, and anemia), and certain pharmaceuticals (cholestyramine, methotrexate, oral contraceptives, phenytoin, carbamazepine, and metformin) [1].

Homocysteine homeostasis is maintained through two primary metabolic pathways: remethylation back to methionine and transsulfuration to cysteine, which also produces hydrogen sulfide (H_2_S). Disruption of these pathways can result in hyperhomocysteinemia (HHcy), typically categorized as mild (15–30 μmol/L), moderate (30–100 μmol/L), or severe (>100 μmol/L). Elevated Hcy levels are associated with an increased risk of neurovascular and neurodegenerative diseases, ischemic injury, dementia, migraines, and epilepsy [2]. The proposed mechanisms of Hcy toxicity include oxidative stress, DNA damage, protein homocysteinylation, apoptosis, and excitotoxicity. Furthermore, Hcy-induced inflammation related to HHcy involves the release of pro-inflammatory cytokines and alterations in DNA methylation, while decreased H_2_S levels may further potentiate the neurotoxic effects of Hcy.

Biochemically, HHcy is characterized by high levels of Hcy and its reactive metabolites, such as Hcy-thiolactone and N-homocysteinylated proteins. Methionine, derived from dietary protein, is metabolized via conversion to S-adenosylmethionine (SAM), a major methyl donor for cellular processes. After methyl group transfer, SAM is demethylated to S-adenosylhomocysteine (SAH), which is then hydrolyzed to homocysteine. Hcy is subsequently remethylated to regenerate methionine or enters the transsulfuration pathway, producing cystathionine. A small fraction of Hcy undergoes intramolecular cyclization, forming homocysteine thiolactone, a reactive intermediate that can modify proteins via N-homocysteinylation. Hcy-thiolactone is generated through an error-editing reaction during protein synthesis and can chemically alter proteins by attaching to lysine residues, potentially forming immunogenic and cytotoxic N-homocysteinylated proteins [3]. The purpose of this paper is to review the data from the specialized literature that demonstrate that there is a direct link between endothelial injury and increased homocysteine levels, identifying existing evidence, describing new mechanisms, and exploring potential new therapeutic options.

## 2. Homocysteine Metabolism and Biochemistry

Homocysteine is a metabolic intermediate in the pathway that converts methionine (Met) to cysteine (Cys). In humans, Hcy is entirely derived from dietary methionine, which supports protein synthesis and the production of S-adenosylmethionine (AdoMet), a universal methyl donor. AdoMet is then converted by demethylation into S-adenosylhomocysteine (AdoHcy), the direct precursor of Hcy [4].

Homocysteine levels are regulated through two primary pathways: remethylation to Met, catalyzed by methionine synthase and betaine–homocysteine methyltransferase (BHMT), and transsulfuration to cysteine, mediated by the enzymes cystathionine β-synthase (CBS) and cystathionine γ-lyase (Figure 1).

Methionine synthase is involved in the remethylation pathway, converting homocysteine to methionine, which is required for protein synthesis and various biological pathways. Vitamin B12 is a cofactor in this reaction in which 5-methyltetrahydrofolate, derived from 5,10-methylenetetrahydrofolate via methylenetetrahydrofolate reductase (MTHFR), donates a methyl group. In the transsulfuration pathway, homocysteine is converted to cysteine by the cystathionine β-synthase (CBS) enzyme, with vitamin B6 as a cofactor. Cysteine is used to produce the powerful antioxidant glutathione in the body [4].

Homocysteine metabolism is affected by several factors that lead to variation in its blood concentration. Enzyme deficiencies in homocysteine metabolism result in elevated homocysteine levels due to genetic mutations affecting the function of these enzymes. A classic example is the MTHFR gene polymorphism (e.g., C677T), which inhibits remethylation of homocysteine and is associated with elevated levels of homocysteine and increased cardiovascular risk. Environmental factors, particularly diet, are also significant. Dietary intake of vitamin B, including vitamin B6, vitamin B12, and folic acid, is essential, along with staying within the normal limits of homocysteine levels in the body. Hyperhomocysteinemia can also result from vitamin deficiencies; moreover, external causes like chronic stress, kidney and liver diseases, and a sedentary lifestyle have been proven to elevate levels of homocysteine [4].

A combination of both preventative and intervention strategies, focusing on genetic components and environmental factors, can reduce the risks associated with hyperhomocysteinemia. Hcy is not used directly in protein synthesis but can post-translationally modify proteins. After being activated by methionyl-tRNA synthetase in an error-editing reaction, Hcy generates Hcy-thiolactone that can bond with lysine residues in proteins; this process leads to the emergence of N-homocysteinylated proteins [5]. Human umbilical vein endothelial cells (HUVECs), the most widely used vascular cell model thus far, have been shown to possess Hcy-thiolactone hydrolyzing activity. In HUVECs incubated in streptomycin-supplemented media [6], both methionine and homocysteine were processed into Hcy-thiolactone using radiolabeled [35S]Met or [35S]Hcy as precursors. These then reacted with lysine residues on proteins to generate N-Hcy-protein [7]. Significantly, folic acid supplementation in the culture medium (M199) prevented the methionine from being converted to homocysteine and Hcy-thiolactone. The attachment to protein through isopeptide bonds was confirmed by the sensitivity of N-Hcy-protein to Edman degradation, a technique that removes N-linked amino acids from proteins. Higher Hcy concentrations resulted in the accumulation of Hcy-thiolactone, which was alleviated by non-radioactive methionine, indicating that its synthesis correlates with that at the methionyl-tRNA synthetase (MetRS) active site. N-Hcy-protein levels also increased with Hcy levels but were inhibited by folic acid and high-density lipoprotein (HDL) supplementation [8]. This highlights the function of folates in the remethylation of homocysteine to methionine [9] and the contribution of paraoxonase 1 (PON1), a component of HDL, in the hydrolysis of Hcy-thiolactone [10].

AdoHcy, the only known biological precursor of Hcy, is generated from AdoMet via reversible hydrolysis by adenosylhomocysteinase (AHCY) [11,12]. Metabolites, including Hcy, Hcy-thiolactone, and N-Hcy-protein, can upregulate AHCY expression [13], which in turn can affect intracellular AdoHcy levels. Since AdoHcy accumulation inhibits methylation reactions, AdoHcy elevation has been linked to increased cardiovascular and cerebrovascular risk, and AdoHcy has been suggested as being a more reliable vascular disease biomarker than Hcy [14].

Youssef-Saliba et al. theorized that homocysteine may have been important in the early stages of protein synthesis during the origin of life [15]. Even if it is not seen as a proteinogenic amino acid anymore, homocysteine is still important in methionine and cysteine metabolism. Hcy-thiolactone is formed from homocysteine when ribosomal editing prevents its wrongful incorporation into peptide chains. They were able to show spontaneous Hcy cyclization in aqueous conditions, as well as its ability to react with amino acids to form dipeptides—hypothesized as potential prebiotic building blocks. Nieraad et al. reviewed the one-carbon (C1) metabolism, the kinetic behavior of enzymes involved, and the common disruptions that involved B-vitamin deficiencies (notably B9, B2, B6, and B12), heterozygous forms of the genetic polymorphisms of the MTHFR and CBS genes, oxidative stress, and lifestyle factors [16].

## 3. Vascular Endothelium Function—Between Normal and Pathologic

Endothelial cells (ECs) are located at the interface between the circulating blood or lymph and the walls of the entire circulatory system, from the heart to the smallest capillaries [17].

ECs have distinct histological features and a strong interrelationship with surrounding tissue through a thin basal lamina. The subendothelial connective tissue contains myointimal cells and scattered fibroblasts, which are essential for the formation of the extracellular matrix that coordinates EC function. Special attention is paid to the physiological and pathophysiological aspects of endothelial cells stemming from this complex interaction [18].

The vascular endothelium is highly distributed in the human body (approximately 1 to 6 × 10^13^ cells), consists of a single layer of squamous cells, which are very flat (about 0.1–2 µm thick, 10–20 µm in diameter), and weighs approximately 1 kg [18,19].

ECs are long and polygonal-shaped with flat nuclei which are aligned in the direction of blood flow and a few organelles, mostly concentrated in the perinuclear zone [18]. The cytoplasm contains actin, myosin, and tropomyosin, as well as pinocytotic vesicles, located adjacent to the endothelial cell membranes. These contractile proteins control motor activities, while the small vesicles are vital for passing materials into and out of a tissue from the blood stream to the underlying tissues [20].

ECs are connected to each other by tight junctions (continuous endothelium). These intercellular junctions are crucial to the integrity of the vessel and selectively limit the movement of macromolecules [21]. In fact, depending on localization and function, the endothelium can be classified into two other structural types: fenestrated (containing pores) and discontinuous (containing gaps). The ECs are responsible for the passage of large molecules or small solutes through continuous and fenestrated endothelium, whereas discontinuous endothelium is highly permeable. These structural variations reflect the complex functional heterogeneity of ECs [18].

Assessment of endothelial function is performed using numerous histochemical and biological techniques. Over time, these tools have provided valuable ultrastructural, histological, and molecular information. The methods include (i) electron microscopy; (ii) histochemistry; and (iii) cytochemistry (CD34, CD31, endothelin, von Willebrand factor). ECs exhibit specific cellular components such as microfilaments, microvilli, caveolae (plasmalemmal vesicles), and Weibel–Palade bodies (containing the von Willebrand factor) [18,22]. Some ECs are surrounded by numerous supporting pericytes, each containing its own basement membrane.

The normal endothelium, considered the innermost side of blood vessels, works as a semipermeable layer and behaves as a sensor and transmitter of the signal in the circulating environment. ECs are important in maintaining the homeostatic balance of the vessels during the production of factors that regulate intracellular signaling, vascular tone, coagulation, neutrophil recruitment, lipid transport, hormone trafficking, and proliferative cellular response, among others [23].

In this context, ECs have numerous arrays of functional and adaptive qualities. These cells can release a wide variety of bioactive substances by acting as agonists or antagonists in response to pathophysiological stimuli [24]. Thus, they produce (i) vasodilatory agents (nitric oxide (NO) and prostacyclin (PGI2)) and vasoconstrictor molecules (endothelin-1 (ET-1), angiotensin-converting enzyme AII (AII), and thromboxan A2 (TxA2)); (ii) procoagulants (protease-activated receptors (PARs), TxA2, plasminogen, activator inhibitor-1, and von Willebrand factor) and anticoagulants (thrombomodulin, heparan, endothelial protein C receptor (EPCR), tissue-type plasminogen activator (tPA), ecto-ADPase, and prostacyclin) [25]; (iii) inflammatory and anti-inflammatory substances; (iv) profibrinolytic and fibrinolytics; and (v) oxidants and antioxidants (Figure 2) [26,27].

The ECs are highly metabolically active and perform their functions in multiple ways, sometimes simultaneously. First, endothelial cells can act as a barrier with selective permeability between blood and tissues, maintain a non-thrombogenic surface, and regulate vascular tone by synthesizing and releasing vasoactive substances. In addition, endothelial cells are a source of (i) cytokines and molecules that regulate cell growth and proliferation; and (ii) adhesion molecules and chemotactic factors that control the migration of leukocytes from the vascular wall (Figure 3).

In accordance with this notion, it is important to mention that endothelial cells can undergo extensive modifications, in particular in the context of hypoxia by disrupting redox balance (producing ROS), or during endoplasmic reticulum stress, in mitochondrial dysfunction, upregulations of enzymes or adhesion molecules, and discharge of proinflammatory factors (producing chemokines) [29,30].

Related to this matter, in 1980 Furchgott and Zawadzki [31] described an endothelium-derived factor that induces arterial relaxation in response to acetylcholine, later identified as NO [32].

Beyond its vasodilating role, NO has antiatherogenic properties, inhibiting platelet aggregation and adhesion, hindering smooth muscle cell proliferation, as well as having an essential role in vascular permeability and inflammatory processes [33].

The classic NO bioactivity pattern involves the binding of the NO to the heme group originating from guanylated cyclase in target cells (platelets, smooth muscle cells) with the intention of increasing cellular cGMP and activating cGMP-dependent protein kinase, thereby influencing NO-mediated vasodilation and platelet inhibition [34,35]. The main pathways involved in vascular tone regulation are schematically summarized in Figure 4.

In this respect, any deficiency of this endothelium-derived vascular regulator with antiadhesive, vasodilatory, and pro-angiogenic effects can change the integrity of the endothelial cell, with many adverse chemical reactions and biological consequences.

Over the last decade it has been observed that endothelial cells are responsible for the synthesis of hydrogen sulfide (H_2_S) that appears to act in a similar manner to NO. Mostly, endogenous H_2_S is produced via enzymatic desulfhydration using cysteine as the substrate. Its synthesis is catalyzed and maintained by the interaction of three enzymes: cystathionine b-synthase (CBS), 3-mercaptopyruvate sulfurtransferase (3-MST), and cystathionine γ-lyase (CSE) [27,36]. H_2_S-induced vasorelaxation may be endothelium-dependent or can act in an independent manner [37]. H_2_S can increase cyclic guanosine monophosphate (cGMP) levels by inhibiting phosphodiesterase A5 (PDE), the enzyme that is involved in its catabolism. The activation of Nox (NADPH oxidase) generates hydrogen peroxide, causing oxidative stress, and stimulates the vasodilatatory-H_2_S-dependent pathway. Also, H_2_S has been shown to relax blood vessels by the involvement of vascular smooth muscle ATP-sensitive K+ (KATP) channels. It is also reported that KATP channels in H_2_S-induced vasodilation work independently of the endothelium [38]. Many research groups have shown that several key mechanisms promote H_2_S-induced vasorelaxation: (i) Cl^−^/HCO_3_^−^ channels [39]; (ii) transient receptor potential (TRP) channels [40]; (iii) Ca^2+^ channels or sparks [41]; (iv) the NO pathway [42]; (v) phospholipase A2 [43]; and metabolic/mitochondrial effects [39]. The impaired H_2_S bioavailability is postulated to be associated with endothelial dysfunction [27].

Regarding the antithrombotic role of the endothelium, it is manifested by the presence of the glycocalyx (negatively charged protein), synthesis of platelet aggregation inhibitors (PGI2, NO), synthesis of antithrombotic proteins (antithrombin III, thrombomodulin), and synthesis of tPA (tissue plasminogen activator) [44].

A series of evidence points to the existence of a third vasodilator mediated by endothelium-derived hyperpolarizing factor (EDHF), which primarily controls relaxation of the microvasculature, alongside NO and prostacyclin, which predominantly control relaxation of the macrovasculature [45,46,47].

EDHF is a major determinant of vascular tone in small resistance vessels. EDHF is synthesized and released from the endothelium under variable physiopathological signaling. EDHF induces endothelium-dependent vascular hyperpolarization and smooth muscle cell (SMC) relaxation of the microvasculature by opening calcium-dependent potassium channels (KCa) because of the decrease in calcium inflow [48,49]. Calcium-dependent potassium channels include three subtypes: (i) small conductance KCa (SKCa1, SKCa2, SKCa3); (ii) intermediate conductance KCa (IKCa); and (iii) large KCa conductance (BKCa) [50]. VSMCs are responsible for releasing large conductance KCa (BKCa) subtypes, while endothelial cells, due to increasing intracellular calcium concentration, release small conductance KCa (SKCa) and intermediate conductance KCa (IKCa) [51]. Under increasing calcium concentration, SKCa and IKCa channels release K+ ions into the subendothelial space, mediating the hyperpolarization and relaxation of vascular smooth muscle [52].

Although extensively studied, the nature of EDHF remains unclear. In general, EDHF-induced endothelium-dependent vascular relaxation is assessed by monitoring the response to an endothelium-dependent agonist during the combined blockade of NO synthases (NOSs) and cyclooxygenases (COXs). In small resistance vessels, the EDHF pathway appears to be even more important than NO in mediating endothelium-dependent vasodilation, serving as a backup mechanism that compensates for reduced NO.

For a better understanding of the role of EDHF, hyper-reactive oxygen species have also been taken into consideration. Cu, Zn-SOD endothelial superoxide dismutase plays an important role in the EDHF component of endothelium-dependent relaxation in mesenteric arteries in both humans and mice [53].

Hydrogen peroxide produced by Cu, endothelial Zn-SOD, works as EDHF in human and mouse mesenteric arteries and in porcine coronary microvessels, having an important protective role in coronary self-regulation and in vivo ischemia/reperfusion lesions [54]. In renal arteries, acetylcholine stimulation activates NAD (P) H oxidase with the release of H_2_O_2_ as a vasoconstrictor derived from endothelium [55].

In 2013, Tang et al. demonstrated that H_2_S may be a major EDHF regulating endothelial function in microvasculature [56]. So, EDHF regulates endothelial function in microvasculature through H_2_S [56]. Garcia et al., and Hoopes et al., reported that 20-HETE (20-hydroxyeicosatetraenoic acid) and EETs (epoxygenase-derived arachidonic acid metabolites) have vascular protective effects [57,58]. Known as vasodilators in microcirculation, 20-HETE (CYP4A derivative) and EETs (CYP2C derivate) have been involved in salt and water secretion and reabsorption, governing in this way the vascular hemodynamics [57,58,59]. The CYP family elicits the activation of KCa channels, relaxing vascular smooth muscle cells and inducing vascular dilation [51,60]. The CYP epoxygenases (CYP2J2, CYP2C8) produce four EET regioisomers: (i) EET-5,6 (mediates renal vasodilation); (ii) EET-8,9 (anti-platelet aggregation effects); (iii) EET-11,12 (pro-fibrinolitic effects) [61]; and (iv) EET-14,15 (decreases inflammation by anti-inflammation effects) [62].

Endothelium-derived EET serves as an EDHF, eliciting hyperpolarization and vascular relaxation by increasing intracellular calcium concentrations and activating large-conductance calcium-sensitive potassium channels in the smooth muscle membrane [51]. Vascular relaxation in response to EETs has been observed in the human microvessels of skeletal muscle, intestines, heart, kidney, and brain [63]. Regarding the possible influence of resting stress levels on the mechanisms involved in endothelium-dependent relaxation, no studies have tackled the issue.

Some studies have been published regarding the contribution of the mechanisms involved in endothelium-dependent relaxation in relation to the contracting agent, as well as an interesting discovery about the K^+^ channel target, focusing on NO and EDHF and their involvement in the precontraction mechanism [64]. Based on previous studies regarding resting tension and vascular smooth muscle contraction, researchers have begun to unravel the influence of resting tension on the endothelium-dependent relaxation profile [65].

Several studies published in the field of endothelium-dependent vasodilation refer mainly to the role and mechanism of EDHF. Thus, in rat resistance mesenteric arteries, H_2_O_2_ has an endothelium-independent relaxing effect mediated by potassium channels with known implications in endothelium-dependent vasodilation [66]. Ascorbic acid has an inhibitory effect on EDHF, but only distally, not in the mesenteric arch [67].

## 4. Endothelial Dysfunction and Cardiovascular Disease

### 4.1. Endothelial Dysfunction: General Principles

Endothelial dysfunction is a general term which implies a decrease in NO synthesis and/or an imbalance between relaxation factors and those of endothelial contraction.

Endothelial dysfunction is reflected primarily in the complex regulations of the main physiological and pathophysiological processes. Morphofunctional changes in endothelial cells cause alterations in all tissues and organs, so it is very difficult to determine or establish the lesional transitions (which of these changes are due to one another). The structural pathological changes identified in the endothelium are not always followed by clinical disorders [18]. On the other hand, important symptoms can be found without morphological substrate. Such discrepancies reflect the intervention of various mechanisms of compensation achieved at molecular level. It is generally accepted that many factors have been suggested to contribute to the pathogenesis of cardiovascular disease or other hemodynamic and metabolic dysfunction prior to any visible morphological changes in vascular endothelium [59].

In response to much pathophysiological stimulation, endothelial cells become prone to develop vascular abnormalities, characterized by vascular endothelium dysfunction and subsequent impairment of endothelium-dependent vasodilation. In addition to hemodynamic dysfunction, several other molecular mechanisms have been involved in the initiation and progression of endothelial cells dysfunction [59].

Activated endothelium becomes a source of inflammatory cells (ROS, NF-kB) and growth factors (ET-1, AII, PDGF, interleukins, FGF); it intervenes in the recruitment of adhesion molecules (CAMs, selectins); it predisposes to thrombosis (PAI-1, TF, TxA2) and vasoconstriction (ROS, ET-1, TxA2, AII) [59]. Damage to the endothelial lining is one of the earliest events in the recruitment of inflammatory cells to the vessel, and in the initiation of immune signaling mediated by T lymphocytes and NK cells [23]. All these unfavorable events promote the migration, proliferation, and permeability of endothelial cells [68].

Impaired endothelial functions could appear independently, without metabolic or inflammatory factors being involved in the alteration of the architecture of the vasculature wall. Moreover, differences in gene expression as well as the genetic background of the patient contribute to the outcome of the patient [69]. Malfunction of endothelium is related to different pathophysiological processes (atherosclerosis, hypertension, thrombosis, stroke, hyperhomocysteinemia, sickle cell anemia) or is associated with increased risk for other diseases (diabetes, ischemia reperfusion injury, neurodegenerative diseases, obesity, rheumatoid arthritis, and severe inflammation response) [70,71,72,73,74]. In addition, losses of endothelial physiological functions are associated with abnormalities in tissue repair, stiffness of vessel walls, inflammation, and sepsis [75].

Current evidence indicates that the effect of impairing NO release can lead to disruption of endothelium in skeletal muscle [76]. In recent years, a considerable number of reports indicate a possible mechanism by which endothelial cells could be altered, including intracellular and biochemical dysfunction, which may alter the basal permeability [77]. According to one study, exposure of altered endothelium to variable endogenous or exogen signaling (stimuli) leads to an increase to macrophage adhesion, stimulating their chemotaxis effect [78]. In this context, in people with rheumatoid arthritis (RA), sedentary behavior sets out a specific vascular outcome that affects the microvascular endothelium-dependent function [74]. Toutouzas et al., 2013, reveal that some RA-related factors could be one of the potential sources of poor microvascular perfusion in the coronary circulation [79].

### 4.2. Cardiovascular Disease in the Context of Endothelial Impairment

Cardiovascular diseases have various etiologies and pathologies. Their complexity is highlighted by the fact that the lesions that characterize them are located mainly on one layer, two layers, or even all three layers. Heart diseases are especially associated with vessel damage. Understanding of these aspects has grown exponentially in the last decade. As in physiological conditions during various states of disease, the heart can adapt to hemodynamic needs, if the myocardium is not harmed or is damaged to a lesser degree [78,79].

It is well known that the endothelium underlies the onset of atherosclerosis, a lesion with a variable combination of changes in the intima of the arteries. Furthermore, it has been observed that endothelial dysfunction, seen as vasomotor dysfunction, occurs before the structural formation of atherosclerosis and is an essential factor in its progression [24]. Atherosclerotic lesions are characterized by focal intimal accumulations of lipids, carbohydrate complexes, blood and blood products, fibrous tissue, and calcium deposits. As a result, the impaired endothelium is directly responsible for reduced myocardial perfusion and myocardial ischemia [20].

The phenomenon is explained by the increased permeability of dysfunctional endothelial cells to LDL-cholesterol particles, followed by their oxidation in the arterial intima [80]. Consequently, a series of phospholipids that activate inflammatory and thrombogenic processes are released, as well as growth factors, thus stimulating the proliferation of smooth muscle cells and the production of excess collagen, with the progression of atheroma plaque [81]. However, etiopathogenesis studies have not been able to establish a direct and exclusive causal relationship between a single etiopathogenic factor and endothelial dysfunction. For these reasons, endothelial failure is considered a multi-factorial disease (Figure 5).

As described above, various studies demonstrate the essential role played by endothelial dysfunction in the pathophysiology of various cardiovascular diseases. Microvascular angina and coronary vasospasm are caused by dysfunctional modulation of hypertensive vascular endothelial dysfunction [82,83]. Thus, in stage I of essential hypertension, approximately 60% of patients have inadequate gluteal subcutaneous vasodilation on biopsy [84]. The same phenomenon has been described for type 1 or 2 diabetes [85], coronary artery disease [86], peripheral arterial disease, and renal failure.

Moreover, endothelial abnormalities can be found in asymptomatic subjects at risk for cardiovascular disease, demonstrating its role in the diagnosis of subclinical atherosclerosis, in assessing a medium- or long-term cardiovascular prognosis, and in choosing an optimal therapeutic course [87]. The precipitation of acute vascular events in atherosclerosis involves processes that go beyond the vulnerability and rupture of the atheroma plaque. Several studies confirm that local processes of homeostasis in the arterial wall are abnormal in both patients with obvious atherosclerosis and those with risk factors for atherosclerosis.

As mentioned before, evidence from large epidemiological studies shows hypertensive vascular dysfunction is due to a reduction in NO synthesis with decreased NO bioavailability, associated with increased vascular oxidative stress that triggers inflammatory cells [88]. The stimulation of the molecules responsible for vascular smooth muscle contraction may not only contribute to hypoperfusion but may also determine vasoconstriction and hypoxia leading to alteration of vascular tone, resulting in hypertension [83].

In the last decade, the incidence of diabetes, obesity, and metabolic syndrome increased with age. Many of these factors are related to endothelial dysfunction. Its complications are highly ranked among cardiovascular disease and subsequent complications. Adipokines secreted by adipose cells are also important for processes by which insulin-mediated vasoreactivity might contribute to endothelial dysfunction [71]. In type 2 diabetes, for instance, perivascular adipose tissue triggered by inflammatory cells is essential for regulating the local vascular tone and leads to marked endothelial dysfunction [71]. Activation of these inflammatory cells relies on an adipokine signaling pathway. Inflammatory background in turn contributes to reduced perfusion and hypoxia [89]. As a complication of hyperglycemia, type 1 diabetes is the main driver of endothelial dysfunction determined by metabolic changes [90].

## 5. Mechanisms of Homocysteine-Induced Endothelial Dysfunction—Experimental and Clinical Data

Most of the clinical studies that examine the connection between HHcy and cardiovascular disease risk have only measured total plasma homocysteine (tHcy). tHcy primarily reflects oxidized forms of Hcy bound to proteins and small thiols [91], where free Hcy represents only a minor portion. Nevertheless, tHcy does not account for several key Hcy-related metabolites such as Hcy-thiolactone, N-Hcy-protein, cystathionine, and S-adenosylhomocysteine. Since HHcy alters both tHcy and these metabolites, it is challenging to attribute vascular damage to one specific compound.

Although Hcy-thiolactone, N-Hcy-protein, and cystathionine levels often parallel tHcy variations, AdoHcy does not consistently do so. In CBS-deficiency humans, serum AdoHcy significantly increases only when tHcy exceeds 100 μM [92]; in Cbs−/− mice that exhibit vastly higher plasma tHcy, serum AdoHcy remains constant while hepatic AdoHcy rises markedly [93].

There is increasing evidence regarding the relationship of abnormally high concentrations of plasma total homocysteine, or plasma hyperhomocysteinemia, to endothelial dysfunction, inflammation, and cardiovascular disease [94,95]. For instance, in 2022, Bajic et al. reviewed evidence supporting homocysteine as both a marker and risk factor for cardiovascular diseases (CVDs), including stroke, myocardial infarction, heart failure, cancer, Alzheimer’s disease, and atherosclerosis [96]. High levels of homocysteine lead to endothelial dysfunction by causing damage to the blood vessel walls and altering their properties from anticoagulant to procoagulant. Additionally, homocysteine activates coagulation factors, disrupts the balance between vasodilators and vasoconstrictors, and induces toxic effects through the inhibition of Na^+^ and K^+^-ATPase. This leads to a reduction in gasotransmitter levels (NO, CO, H_2_S), overstimulation of NMDA receptors, and the promotion of inflammation, oxidative stress, and mitochondrial dysfunction in cardiac tissue.

The oral methionine load test (100 mg/kg) is used to detect impaired homocysteine metabolism in clinical studies. Elevated plasma tHcy in response to a single methionine load persists longer in CVD patients than in healthy controls, and an abnormal response is thought to be an independent risk factor for coronary, peripheral, and cerebral vascular disease [97,98]. The detrimental effect of low-dose methionine or dietary animal protein on endothelial function has been documented in otherwise healthy subjects as an increase in plasma tHcy and a decrease in flow-mediated dilation (FMD) [99]. Alternatively, a methionine-free amino acid mixture did not influence the results, indicating that even minor changes in tHcy contribute to endothelial dysfunction and the progression of atherosclerosis.

A study in healthy volunteers showed that only free reduced Hcy was closely linked to endothelial dysfunction, with flow-mediated dilation decreasing and Hcy levels rising after oral Hcy or Met administration. This relationship was most potent when reduced Hcy peaked, while oxidized forms had no consistent effect [100].

In hyperhomocysteinemia, changes occur not only in plasma tHcy levels but also in individual Hcy-related metabolites. A study examined how these metabolites—Hcy, Hcy-thiolactone, and N-Hcy-protein—affect gene expression in HUVEC using microarray, RT-qPCR, and bioinformatics [13]. Each metabolite induced specific alterations in gene expression, contributing to endothelial dysfunction in HHcy. Identifying these metabolite-specific changes could reveal molecular pathways involved in HHcy-related diseases and suggest targets for interventions to treat cardiovascular and neurological disorders.

Several studies have documented higher levels of homocysteine in multiple sclerosis (MS) patients than in healthy controls. Increased homocysteine causes neuronal injury via oxidative stress, excitotoxicity, and apoptosis. It stimulates receptors for glutamate, which raises calcium levels and the production of reactive oxygen species that can damage nerve cells. Excessive homocysteine levels may also destabilize myelin by decreasing S-adenosylmethionine availability. MS patients with higher levels of homocysteine have been associated with faster disease progression, more significant disability, and cognitive impairments. Furthermore, studies indicate an association between elevated homocysteine levels and worsened clinical outcomes, including elevated Expanded Disability Status Scale (EDSS) scores and a more rapid rate of disease progression [101].

The treatments with N-Hcy-protein, Hcy-thiolactone, and Hcy all altered 21 genes in the same direction. Notably, N-Hcy-protein, Hcy-thiolactone, and Hcy upregulated genes involved in one-carbon metabolism, including CBS, MTR, and MTRR [13]. Key pathways influenced by Hcy-thiolactone included chromatin organization, one-carbon metabolism, and lipid processes, while all three metabolites affected pathways like blood coagulation, lipid metabolism, wound healing, and sulfur amino acid biosynthesis. Atherosclerosis and coronary heart disease were the top diseases linked to all three metabolites. Hcy-thiolactone correlated explicitly with myocardial infarction, a connection confirmed by a large prospective study [102].

These findings suggest that Hcy metabolites uniquely modulate gene expression, contributing to endothelial dysfunction and vascular disease in HHcy [13].

Xiao & colab. clarified how AdoHcy contributes to endothelial dysfunction [103]. While elevated AdoHcy—an immediate precursor of homocysteine—is known to be linked to increased CVD risk and atherosclerosis progression, its direct role in impairing endothelial function has not been fully understood. To investigate this, researchers used Apolipoprotein E-deficient (apoE−/−) mice treated either with adenosine dialdehyde, an inhibitor of AdoHcy hydrolase (AHcy), or injected with a retrovirus carrying Ahcy shRNA. These interventions, along with the use of Ahcy heterozygous knockout mice (Ahcy+/−), led to elevated plasma AdoHcy. Elevated AdoHcy levels in these models were associated with impaired endothelium-dependent relaxation and reduced nitric oxide (NO) bioavailability in response to acetylcholine. This impairment was reversed by the endothelial NO synthase inhibitor N(G)-nitro-L-arginine methyl ester, confirming NO pathway involvement. Moreover, the inhibition of AHcy increased reactive oxygen species (ROS) and upregulated p66shc expression in both mouse and human aortic endothelial cells. The use of antioxidants or p66shc-targeting siRNA reversed ROS generation and improved vascular responses.

Epigenetically, AHcy inhibition resulted in hypomethylation of the p66shc promoter and downregulation of DNA methyltransferase 1 (DNMT1). Overexpression of DNMT1 via adenoviral transduction reduced p66shc levels, supporting a mechanistic link between AHcy inhibition, epigenetic modulation, and oxidative stress. In human patients with coronary artery disease and healthy control groups, plasma AdoHcy levels showed a negative correlation with flow-mediated dilation and promoter methylation of p66shc and a positive correlation with oxidative stress. Overall, these findings suggest that elevated AdoHcy induces endothelial dysfunction by epigenetically activating p66shc-mediated oxidative pathways, thereby contributing to vascular injury and atherosclerosis. Notably, AdoHcy may serve as a more sensitive marker of CVD risk than total homocysteine [103].

In 2021, Prtina et al., in a 118-patient study, assessed the impact of high-dose vitamin D supplementation on homocysteine, vitamin B12, folate, and inflammatory cytokines in patients with plaque psoriasis—a condition characterized by both local and systemic inflammation, often accompanied by elevated Hcy levels. After three months, they observed that there was a significant increase in vitamin B12 levels, along with reductions in both Hcy and folate. These outcomes suggest that vitamin D may influence the regulation of genes involved in Hcy metabolism [104].

Studies have demonstrated that homocysteine triggers inflammatory responses in vitro in endothelial cells [105].

MicroRNAs (miRs) are small non-coding RNAs that regulate gene expression at the mRNA level, and their dysregulation is linked to endothelial dysfunction [106]. Plant homeodomain finger protein 8 (PHF8) is a histone demethylase that regulates mTOR signaling located on the X chromosome. It is associated with intellectual disabilities, autism, and ADHD [107]. In mice with Cbs deficiency and neuroblastoma cells, HHcy downregulated PHF8 expression [108]. mTOR signaling has a central role in the regulation of cell metabolism and cell survival, driving anabolic processes and blocking autophagy in a nutrient-replete environment. In atherosclerosis and CVD, dysregulated mTOR signaling represents a prominent pathway [109,110]. In HUVEC treated with Hcy, Hcy-thiolactone, and N-Hcy-protein, each metabolite downregulated PHF8 expression by upregulating miR-22-3p and miR-1229-3p, which target PHF8′s 3′UTR [108].

A study found that during early pregnancy, higher levels of homocysteine were linked to a greater risk of adverse pregnancy outcomes (APOs) such as preeclampsia, preterm birth, and low birth weight, while lower folate levels appeared to offer some protection. The risk of APOs increased steadily across higher Hcy levels but decreased with higher folate levels. The research also showed that both elevated Hcy and lower folate concentrations—along with whether the mother took folate supplements—were independently associated with the likelihood of APOs. However, no connection was found between vitamin B12 levels and APOs. These results suggest that taking folate early in pregnancy may alleviate the risk of complications [111].

Endothelial autophagy and the involvement of homocysteine is another recently described mechanism. In THP-1 macrophages, hyperhomocysteinemia reduces autophagy markers (LC3-II/I, Beclin-1) and increases p62, indicating impaired autophagosome formation. Mechanistically, homocysteine inhibits AMPK activation while promoting mTOR phosphorylation and nuclear translocation of TFEB—collectively suppressing autophagy via the AMPK–mTOR–TFEB axis [112]. In HUVECs, homocysteine exposure elevates autophagic flux and accelerates endothelial senescence. Zhang & colab. in 2023 explored the relationship between autophagy and HCY-induced HUVEC senescence and confirmed that HCY increased HUVEC senescence and autophagy by inducing the production of intracellular ROS [113]. Homocysteine induces mTORC1 activation in neuronal cells, leading to reduced clearance of toxic proteins such as amyloid β and phospho-Tau. This mechanism was identified in human neurons derived from iPSCs and mouse models, with mTOR inhibition or autophagy induction rescuing neurodegenerative phenotypes [114]. Other recent studies have described a link between homocysteinemia and the impact on iron metabolism. In human umbilical vein endothelial cells (HUVECs), Hcy disrupts Akt signaling, leading to elevated ferritin levels—both light (L) and heavy (H) chains—indicative of altered iron storage and impaired iron availability [115]. In another study, it is mentioned that in HUVECs, Hcy induced intracellular iron deficiency, elevated reactive oxygen species (ROS), mitochondrial dysfunction, and apoptosis [116]. On the other hand, in endothelial cells, Hcy leads to lipid peroxidation and iron accumulation triggered through dysregulation of the system Xc–/GPX4 axis, causing ferroptosis [117]. Taking all this into account, we can say that Hcy disrupts iron metabolism in two contrasting ways—by inducing functional iron deficiency and, in other contexts, triggering iron-driven oxidative stress (ferroptosis). The PGRN/EphA2 axis emerges as a key mediator linking elevated homocysteine to cardiovascular and renal dysfunction. Loss of this pathway under homocysteinemia leads to impaired endothelial health, while PGRN supplementation reverses toxic effects—spotlighting it as a novel therapeutic target [118]. Future research should dissect tissue-specific regulatory mechanisms and evaluate clinical applicability of PGRN-centered therapies.

## 6. Possible Therapeutic Ways to Lower Homocysteine

Most clinical trials have evaluated the potential of lowering total homocysteine via B-vitamin supplementation. Mild HHcy is usually treated with vitamin supplementation—particularly folic acid, B6, and B12. For instance, folic acid (0.2–15 mg/day) has been confirmed to decrease tHcy levels, according to the American Heart Association, yet the outcomes for cardiovascular risk remain unresolved [119]. However, it has shown promise in slowing carotid atherosclerosis in the primary prevention of stroke and also cognitive decline; thus, it is generally favorable given its low risk.

Homocysteine is a marker of total vascular damage that must be monitored and controlled as early as possible. Dietary and lifestyle changes are recommended for most patients with hyperhomocysteinemia. Chronic deficiency of folate and vitamin B12 are widely recognized as common contributors to HHcy, and adequate supplementation of these vitamins can significantly lower Hcy levels. Nutrition that includes foods rich in folate and B12, such as fruits, vegetables, and low-fat dairy, is recommended. Methionine ingestion should also be lowered in order to control Hcy levels [120,121].

Recent studies have focused on ameliorating Hcy-induced endothelial injury. For example, melatonin protects endothelial cells due to reducing oxidative stress. It lowers levels of reactive oxygen species (ROS) and lipid peroxidation induced by Hcy, providing antioxidant effects and modulating pro- and anti-apoptotic proteins to inhibit endothelial cell apoptosis [122]. Other compounds under investigation include propofol, glucagon-like peptide 1 (GLP-1) analogs, epigallocatechin gallate (EGCG), estrogen, nicorandil, and L-cystathionine. These compounds may exert protective effects by promoting protective protein signaling, reducing inflammation, improving cell viability, and restoring balance to Hcy-mediated endothelial dysfunction [123].

It has been reported that statins, including atorvastatin, protect against Hcy-induced endothelial injury, further supporting the significance of lipid-lowering therapy in managing HHcy and its associated cardiovascular complications [124,125].

Antihomocysteine effects of B-vitamin therapy in clinical trials were limited to a reduction in the concentration of tHcy without significant effects on other homocysteine-related metabolites. In older adults with high tHcy, B-vitamins offered no benefit to the AdoHcy/AdoMet axis, even though tHcy was lowered after two years of supplementation [126]. The absence of significant AdoHcy reductions may explain why B-vitamin therapy failed to deliver a substantial cardiovascular benefit.

Supplementation with B-vitamins had no effect on Hcy-thiolactone levels, another risk factor for myocardial infarction in patients with coronary artery disease [127]. However, experimental models suggest benefits. B6 and folic acid supplementation have been found to improve heart function in myocardial infarction and heart failure models [96].

Management of classical homocystinuria (HCU) begins with consideration of vitamin B6 (pyridoxine) responsiveness, as about a third of patients are vitamin B6 responsive. In non-responders, treatment involves a methionine-free diet and methionine-free essential amino acid formulas, along with folate and B12 supplementation to aid remethylation. Betaine, which facilitates the conversion of homocysteine to methionine, is administered in doses of 100–200 mg/kg/day administered in divided doses. The primary aim is to keep homocysteine levels less than 50 μmol/L in order to reduce vascular and systemic complications, but ocular involvement is still challenging to manage.

In remethylation disorders such as cobalamin C (CblC) deficiency, treatment includes parenteral hydroxocobalamin rather than cyanocobalamin for the best B12 levels. Betaine dosage is adjusted to maintain balanced homocysteine and methionine levels. In contrast to classical HCU, there is no need for protein restriction in remethylation disorders. In these situations, folic acid and carnitine may be considered but have shown limited benefit [128].

There is no doubt that experimental studies indicate further therapeutic possibilities. For example, Yakovleva et al. found that prenatal HHcy-induced oxidative stress and behavioral abnormalities in rats are linked to reduced cystathionine-beta synthase (CBS) and hydrogen sulfide (H_2_S) levels [129].

In Table 1, Table 2 and Table 3, we summarize some of the therapeutic strategies for hyperhomocysteinemia.

## 7. Conclusions

Enhanced levels of blood homocysteine are an integral part of endothelial dysfunction, which promotes vascular damage and is involved in various cardiovascular disorders including atherosclerosis and thrombosis. The depletion of nitric oxide, the apoptosis of endothelial cells, and the oxidative stress produced by homocysteine result in an imbalance of vascular homeostasis. B-vitamin supplementation lowers Hcy levels and prevents endothelial injury, but it does not improve clinical outcomes. High-dose folate may even cause proinflammatory effects that outweigh potential cardiovascular benefits.

Another important outcome of this review is the call for further investigations regarding the impact of Hcy on more global aspects, especially its implications for cognitive dysfunction. In 2022, Nieraad et al. highlighted, based on preclinical studies correlating hHcy with cognitive decline, that there is a need for large clinical trials to clarify if normalizing Hcy would confer therapeutic benefits for cognitive disorders [130].

Nonetheless, and as summed up well by the authors themselves in the accompanying discussion, the existing body of literature indicates that while substantial progress has been made in clarifying the role of Hcy in vascular injury and how to design interventions targeting this pathway, more work is required to refine therapeutic strategies and also to examine the potential implications of Hcy for other clinical endpoints.

## Figures and Tables

**Figure 1 ijms-26-06298-f001:**
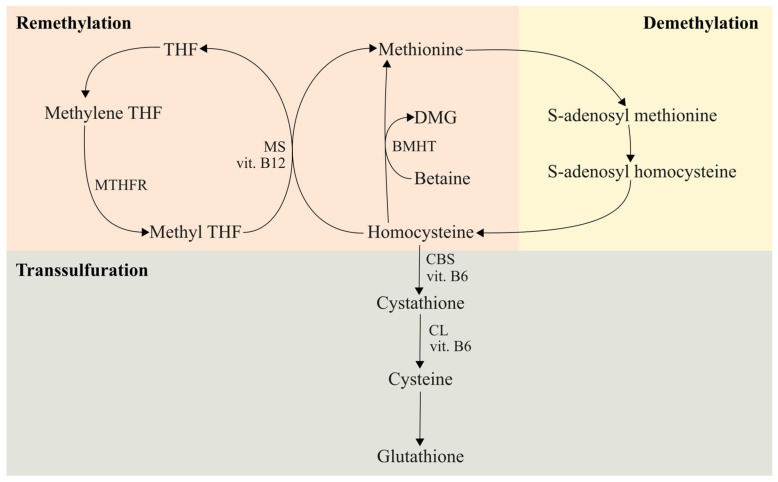
Schematic representation of homocysteine metabolism: remethylation, demethylation, and transsulfuration. THF—tetrahydrofolate, MS—methionine synthase, MTHFR—methylenetetrahydropholate reductase, BMHT—betaine–homocysteine methyltransferase, DMG—dimethyl glycine, CBS—cystathionine β-synthase, CL—cystathionine γ-lyase. The figure was adapted by the authors after Alimov-2024 [4].

**Figure 2 ijms-26-06298-f002:**
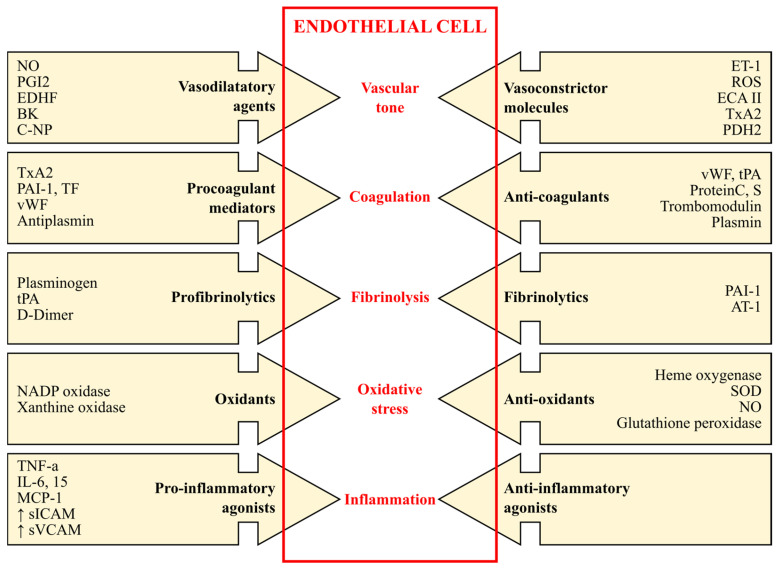
Vascular endothelium function—between normal and pathologic. Nitric oxide—NO; prostacyclin—PGI2; endothelium-derived hyperpolarizing factor—EDHF; bradykinin—BK; C-type natriuretic peptide—C-NP; endothelin-1—ET-1; reactive oxygen species—ROS; angiotensin-converting enzyme AII—ECA II; thromboxan A2—TxA2; prostaglandin H2—PDH2; plasminogen activator inhibitor 1—PAI-1; tisular factor—TF; von Willebrand factor—vWF; tissue activator of plasminogen—tPA; angiotensin II receptor type 1—AT-1; nicotinamide adenine dinucleotide phosphate oxidase—NADPH oxidase; superoxide dismutase—SOD; tumor necrosis factor alpha—TNF-α; interleukin-6—IL-6; interleukin-15—IL 15; monocyte chemotactic protein-1—MCP-1; soluble intercellular adhesion molecule—sICAM; soluble vascular cell adhesion molecule—sVCAM; C-reactive protein—CRP.

**Figure 3 ijms-26-06298-f003:**
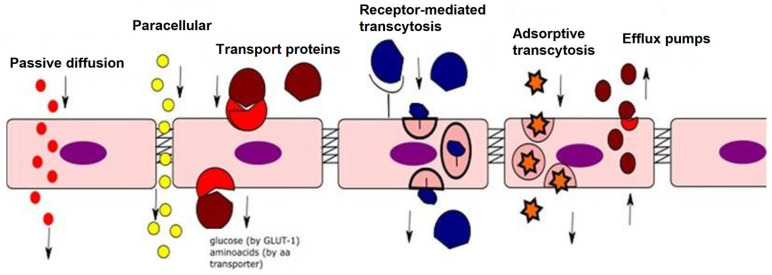
Endothelial cell transport pathways. Red small circles—lipophilic drugs (passive diffusion); Yellow small circles—cations (ex: sodium) through tight junctions (paracellular); Red shape—transport protein (ex GLUT-1 or Glucose Transporter 1, aminoacid transporter) and brown shape—glucose, aminoacids; Blue shape—insulin molecule binding to a cell surface receptor (Pink shape) and transported across the endothelial cell (receptor mediated transcytosis); Starred shape—albumin molecule can be transported in this way (adsorptive transcytosis); Dark red circles—cellular proteins which can actively transport molecules (efflux pumps); Purple ovals—nuclei of endothelial cell.ECs share many molecular pathways and biological mechanisms in a coordinated and cooperative manner [28]. In the context of endothelial cell biology, any imbalance of endothelial cell activation and inflammation, or activation and dysfunction, exerts diverse biological effects by acting on various targeted signaling pathways via multidimensional mechanisms. These responses involve a multitude of clinically relevant effects, persisting until the balance between production and degradation is lost [28].

**Figure 4 ijms-26-06298-f004:**
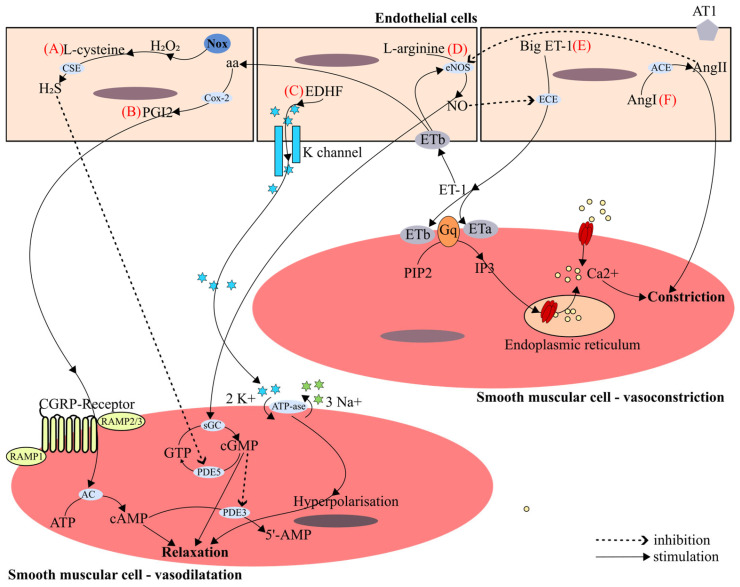
Key mechanisms involved in regulation of vascular tone: (**A**). Hydrogen sulfide (H_2_S) is produced endogenously by direct enzymatic desulfhydration of L-cysteine, catalyzed by cystathionine-γ-lyase (CSE). H_2_S can increase cyclic guanosine monophosphate (cGMP) levels by inhibiting phosphodiesterase A5 (PDE5), the enzyme that is involved in its catabolism. The activation of Nox (NADPH oxidase) generates hydrogen peroxide (H_2_O_2_), causing oxidative stress and stimulating the vasodilatatory-H_2_S-dependent pathway. Low concentrations of cGMP can inhibit phosphodiesterase A3 (PDE3), leading to increased cAMP through decreasing conversion into 5′-AMP, and enhanced barrier function, while higher concentrations of cGMP can activate phosphodiesterase A2 (PDE2), which may increase permeability. (**B**). Prostacyclin (PGI2) is synthesized by cyclooxygenase-2 (Cox-2) from arachidonic acid (aa) and increases the second messenger cyclic adenosine monophosphate (cAMP) in smooth muscle cells. Its action is mediated through specific cell-surface, 7-membrane-spanning, G protein-coupled receptors (CGRP-Receptors) that activate adenylate cyclase (AC), which catalyzes the conversion of adenosine triphosphate (ATP) into cAMP. (**C**). EDHF (endothelium-derived hyperpolarization factor) causes hyperpolarization of the underlying vascular smooth muscle via a mechanism involving increased potassium (K^+^) conductance resulting in a reduction in intracellular K^+^ through ATP-ase pump involved in the exchange of potassium (2K^+^) and sodium (3Na^+^) ions with the subsequent repolarization of the cell and relaxation. (**D**). Nitric oxide (NO) is synthesized by nitric oxide synthase (eNOS) in endothelial cells (ECs) from L-arginine. NO diffuses across biological membranes and stimulates a soluble guanylyl cyclase (sGC), leading to an increase in intracellular cGMP within the adjacent smooth muscle cells (SMCs), thus resulting in their relaxation and consequent vasodilation. (**E**). Endothelin-1 (ET-1) generated by the endothelin-converting enzyme (ECE)-mediated cleavage of a larger precursor (big ET-1). Endothelin receptors Eta and ETb are expressed on vascular SMCs and mediate vasoconstriction. Gq, a type of G protein, plays a crucial role in initiating intracellular calcium release by activating phospholipase C, which hydrolyzes phosphatidylinositol 4,5-bisphosphate (PIP2) into inositol 1,4,5-trisphosphate (IP3) that binds to IP3 receptors on the endoplasmic reticulum, triggering the release of calcium (Ca^2+^) into the cytosol. ETb is also expressed on ECs where its activation results in vasodilation mediated by PGI2 and NO release. (**F**). All components of the RAS (the renin–angiotensin system), except for renin, are produced in the vasculature. Angiotensin-converting enzyme (ACE) converts angiotensin I (Ang I) to active angiotensin II (Ang II). The latter exerts its activities through binding to Ang II type 1 (AT1) or type 2 (AT2) receptors, causing vasoconstriction.

**Figure 5 ijms-26-06298-f005:**
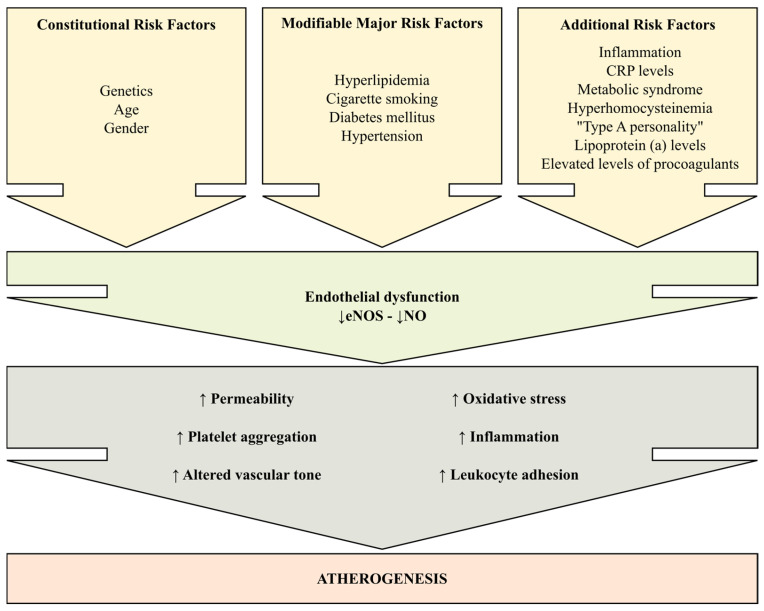
Endothelial dysfunction (diminished production/availability of nitric oxide) is a response to cardiovascular risk factors and precedes the development of atherosclerosis. Endothelial dysfunction is involved in lesion formation by upregulation of adhesion molecules, increased chemokine secretion and leukocyte adherence, increased cell permeability, enhanced low-density lipoprotein oxidation, platelet activation, cytokine elaboration, and vascular smooth muscle cell proliferation and migration.

**Table 1 ijms-26-06298-t001:** Therapeutic schemes in mild to moderate hyperhomocysteinemia.

Treatment	Indication/Effect	References
Folic Acid (0.2–15 mg/day)	Lowers tHcy; may slow carotid atherosclerosis and cognitive decline.	[119]
Vitamin B6 (Pyridoxine)	Lowers tHcy; potential cardiovascular benefits.	[96]
Vitamin B12	Reduces tHcy levels, especially in folate/B12 deficiency.	[120,121]
Dietary Changes	Folate/B12-rich foods; reduce methionine intake to control Hcy.	[120,121]
Statins (e.g., Atorvastatin)	Protects endothelium from Hcy-induced damage; supports lipid-lowering.	[124,125]
Melatonin	Antioxidant effects; protects against endothelial injury from Hcy.	[122]
GLP-1 Analogs, EGCG, Estrogen, Nicorandil, L-cystathionine	Experimental agents that reduce oxidative stress and improve endothelial function.	[123]

**Table 2 ijms-26-06298-t002:** Therapeutic schemes in classical homocystinuria.

Treatment	Indication/Effect	References
Vitamin B6 (Pyridoxine)	First-line in B6-responsive patients (~⅓ cases).	[96]
Methionine-Free Diet + Special Amino Acid Formula	Used in B6 non-responders to reduce Hcy levels.	[128]
Folate + Vitamin B12	Support remethylation of Hcy to methionine.	[120,128]
Betaine (100–200 mg/kg/day)	Converts Hcy to methionine; lowers tHcy < 50 µmol/L.	[128]

**Table 3 ijms-26-06298-t003:** Therapeutic schemes in remethylation disorders (e.g., cobalamin C—CblC deficiency).

Treatment	Indication/Effect	References
Parenteral Hydroxocobalamin	Preferred over cyanocobalamin; improves B12 status and Hcy control.	[128]
Betaine	Balances Hcy and methionine levels.	[128]
Folate	May support remethylation; limited benefit.	[128]
Carnitine	Sometimes used; benefits are unclear.	[128]

## Data Availability

No new data were created or analyzed in this study.

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
