# Peer review of "Homocysteine Attack on Vascular Endothelium—Old and New Features"

_ijms, 2025, doi:10.3390/ijms26136298_

Round 1
Reviewer 1 Report
Comments and Suggestions for Authors
This manuscript lacks clearly defined objectives. The review does not effectively support the central theme of Hyperhomocysteinemia. In my opinion, the topic is neither original nor does it present new insights or directions. Additionally, the review appears to offer limited contribution to the existing body of research on homocysteinemia and its role in cardiovascular disease. The information presented is primarily derived from previously published materials, with insufficient demonstration of connections between studies. Consequently, after careful review, I am not convinced that further revisions would enhance the manuscript or add valuable insights.
Comments on the Quality of English LanguageWhile the language quality is acceptable, there are multiple typographical errors.
Reviewer 2 Report
Comments and Suggestions for Authors
This manuscript provides a broad and ambitious review of the pathophysiological role of homocysteine (Hcy) in vascular endothelial dysfunction and its implications in cardiovascular and neurological disorders. It covers the biochemical metabolism of homocysteine, its pathological effects (including oxidative stress, inflammation, and epigenetic modulation). It also discusses therapeutic strategies targeting hyperhomocysteinemia (HHcy). While the manuscript is generally well-referenced and contains valuable scientific information, it suffers from several critical issues that require substantial revision
The authors should narrow the scope to emphasize the vascular effects of homocysteine, while briefly acknowledging other systemic implications where appropriate.
Numerous paragraphs, especially those describing basic vascular biology and homocysteine metabolism, lack proper referencing. Many statements are presented without citation, undermining the scientific credibility of the review. Furthermore, the authors frequently cite outdated review articles, some published over two decades ago, rather than referring to original research or recent primary literature.
Many sections, especially on endothelial biology, repeat basic information already covered earlier (e.g., repeated explanations of NO, EDHF, ROS, and vascular tone regulation). Substantially condense the description of basic endothelial physiology, and emphasize novel or underexplored insights into homocysteine’s effects.
Numerous grammatical errors (e.g., "vascular endothelium dysfunction" instead of "endothelial dysfunction") and awkward phrases (e.g., "have been suggested to contribute") detract from readability and professionalism. The manuscript requires a thorough language edit by a native or fluent academic English speaker.
A more balanced analysis of the evidence, including a discussion of why interventions lowering tHcy often fail to improve clinical outcomes, would enhance the review’s credibility.
Figure 4 contains several inaccuracies and conceptual oversimplifications:
- Nitric oxide (NO) activates soluble guanylate cyclase (sGC), not simply “GC”.
- The figure implies that phosphodiesterase (PDE) activity is limited to cGMP, omitting its well-established role in cAMP hydrolysis.
- No PDE subtype is identified, although this is crucial given the specificity of PDE isoforms (e.g., PDE5 for cGMP, PDE3 for cAMP).
The entire figure needs to be rethought and redesigned to meet the standards of clarity and scientific accuracy expected in a high-impact journal.
Author Response
Please see in attachment.

Round 2
Reviewer 1 Report
Comments and Suggestions for Authors
Overall, the manuscript has improved. I would like to suggest a modification to the background color of Figure 4, as the current color appears quite bright; a lighter shade would be preferable. Additionally, the outlines in the flow charts (Figures 2, 4, and 5) are dotted; changing these to solid lines would enhance the visual clarity.
Author Response
"Please see the attachment."

Reviewer 2 Report
Comments and Suggestions for Authors
The revised version of the manuscript addresses the major concerns raised in the initial review. The authors have narrowed the focus, updated the references with more recent and primary literature, improved the discussion regarding therapeutic limitations, and revised the language for better clarity. While minor issues such as overly didactic passages and incomplete labeling in Figure 4 remain, these do not compromise the overall scientific merit of the review.
Author Response
"Please see the attachment."
